# Risk factors and high-risk subgroups of severe acute maternal morbidity in twin pregnancy: A population-based study

Diane Korb[1,2]*, Thomas Schmitz[1,2], Aurélien Seco[1,3], François Goffinet[1,4], Catherine Deneux-Tharaux[1], for the JUmeaux MODe d'Accouchement (JUMODA) study group and the Groupe de Recherche en Obstétrique et Gynécologie (GROG)

**1** Université de Paris, Epidemiology and Statistics Research Center/CRESS, INSERM, INRA, Paris, France, **2** Department of Obstetrics and Gynecology, Robert Debré Hospital, APHP, Paris, France, **3** Clinical Research Unit of Paris Descartes Necker Cochin, APHP, Paris, France, **4** Port-Royal Maternity Unit, Cochin Hospital, APHP, Paris, France

* diane.korb@inserm.fr

## Abstract

### Objective

To determine risk factors of severe acute maternal morbidity in women with twin pregnancies and identify subgroups at high risk.

### Methods

In a prospective, population-based study of twin deliveries, the JUMODA cohort, all women with twin pregnancies at or after 22 weeks of gestation were recruited in 176 French hospitals. Severe acute maternal morbidity was a composite criterion. We determined its risk factors by multilevel multivariate Poisson regression modeling and identified high-risk subgroups by classification and regression tree (CART) analysis, in two steps: first considering only characteristics known at the beginning of pregnancy and then adding factors arising during its course.

### Results

Among the 8,823 women with twin pregnancies, 542 (6.1%, 95% confidence interval (CI) 5.6–6.6) developed severe acute maternal morbidity.

Risk factors for severe maternal morbidity identified at the beginning of pregnancy were maternal birth in sub-Saharan Africa (adjusted relative risk (aRR) 1.6, 95% CI 1.1–2.3), pre-existing insulin-treated diabetes (aRR 2.2, 95% CI 1.1–4.4), nulliparity (aRR 1.6, 95% CI 1.3–2.0), IVF with autologous oocytes (aRR, 1.3, 95% CI, 1.0–1.6), and oocyte donation (aRR 2.0, 95% CI 1.4–2.8); CART analysis identified nulliparous women with oocyte donation as the subgroup at highest risk (SAMM rate: 14.7%, 95% CI, 10.3–19.1).

At the end of pregnancy, additional risk factors identified were placenta praevia (aRR 3.5, 95% CI 2.3–5.3), non-severe preeclampsia (aRR 2.5, 95% CI 1.9–3.2), and macrosomia for either twin (aRR 1.7, 95% CI 1.3–2.1); CART analysis identified women with both oocyte

**Data Availability Statement:** The data underlying the findings cannot be made freely available because of ethical and legal restrictions. This is because the present study includes an important number of variables that, together, could be used to re-identify the participants based on a few key characteristics and then be used to have access to other personal data. Therefore, the French National Data Safety Authority (CNIL) strictly forbids making such data freely available. However, they can be obtained upon request from the JUMODA steering committee. Readers may contact: diane. korb@inserm.fr or epope@inserm.fr to request the data.

**Funding:** Thomas Schmitz was supported by a grant from the French Ministry of Health (Programme Hospitalier de Recherche Clinique, AOM2012). The funder had no role in study design, data collection and analysis, decision to publish, or preparation of the manuscript.

**Competing interests:** The authors have declared that no competing interests exist.

donation and non-severe preeclampsia (SAMM rate: 28.9%, 95% CI, 19.9–37.9) and sub-Saharan nulliparous women with non-severe preeclampsia (SAMM rate: 26.9%, 95% CI, 9.9–43.9) as the two subgroups at highest risk.

## Conclusion

In woman with twin pregnancy, rates of severe acute maternal morbidity vary between subgroups from 4.6% to 14.7% and from 3.8% to 28.9% at the beginning and at the end of pregnancy respectively, depending on the combined presence of risk factors.

## Introduction

Twin pregnancies account for about 3% of all births in the United States and France [1,2]. Compared with women with singleton pregnancies, women with twin pregnancies have a fourfold increased risk of developing severe acute maternal complications, mainly during the intra and postpartum periods [3]. However this overall risk augmentation may actually vary by subgroups of women.

Risk factors of severe acute maternal morbidity in women carrying twins have been poorly characterized because previous studies, sparse and old, used variable definitions of maternal morbidity or lacked a specific control group, adequate sample size, and individual data for adjustment [4,5]. Moreover, extrapolation to twin pregnancies of risk factors identified in singleton ones might be inappropriate. Indeed, since twin pregnancy itself constitutes a high risk clinical context, its presence could modify the profile of other risk factors. Therefore, studies of the risk factors of severe acute maternal morbidity in the specific population of twin pregnancies are needed. Understanding which subgroups are at highest risk of severe acute maternal morbidity would be useful in both counseling women and for clinicians to be alert for early recognition and treatment should an event occur to limit its severity.

Because the JUmeaux MODe d'Accouchement (JUMODA) prospective cohort collected detailed individual data in a large population of twin pregnancies, it offered the opportunity to determine risk factors of severe acute maternal morbidity in women with twin pregnancies and identify subgroups at high risk [6].

## Materials and methods

The JUMODA national, observational, prospective, population-based cohort study of twin deliveries took place in France from February 10, 2014, through March 1, 2015 [6]. All French maternity units performing more than 1500 annual deliveries were invited to participate, regardless of their academic, public, or private status or level of care, and 176 of the 191 eligible units (92%) agreed. Women who gave birth at or after 22 weeks of gestation were included (n = 8823). Enrolment took place prospectively immediately after delivery.

Detailed information about the participating women and maternity units has been already reported elsewhere [6]. Research nurses collected data about maternal characteristics, medical history, pregnancy complications, maternal complications, neonatal health, and maternity unit characteristics.

The primary outcome was a composite of severe acute maternal morbidity. This multicriteria definition was developed in a formal national Delphi expert consensus process for another study specifically conducted to define and study it (S1 Table) [3,7]. To include

conditions involving severe health impairments, it combined diagnoses, organ dysfunctions, and interventions, as recommended by WHO [8]. Severe acute maternal morbidity was therefore defined as one or more of the following: maternal death; severe haemorrhage, defined by need for second line therapy, transfusion $\geq$ 4 units of packed red blood cells, uterine artery embolization, vascular ligation, compressive uterine suture, emergency peripartum hysterectomy; eclampsia; preeclampsia responsible for induced preterm delivery before 32 gestational weeks mainly for the mother's health; pulmonary embolism; stroke or cerebral transient ischemic attack; severe psychiatric disorder; cardiovascular or respiratory dysfunction, renal dysfunction (creatininemia >1.47 mg/dL or oliguria <500 mL/24 h), neurological dysfunction (coma of any stage and duration), or hematological dysfunction (thrombocytopenia <50 000/ mm$^3$ or acute anemia <7 g/dL, in the absence of a chronic disorder); emergency surgery in addition to the childbirth procedure, e.g., secondary hysterectomy, laparotomy for a post-delivery complication other than hematoma or wound infection; admission to an intensive care unit (ICU). This primary outcome was treated as a binary variable.

The incidence of severe acute maternal morbidity was calculated, with its 95% confidence interval, as the number of women with a severe acute maternal morbidity event, divided by the total number of pregnancies ending in still- or live birth at or after 22 weeks of gestation in the JUMODA cohort. Among women with such an event, we described the distribution of the underlying causal conditions.

The characteristics of women and of their pregnancies that we tested as potential risk factors for severe acute maternal morbidity were selected from the literature and analyzed in two steps. First, we included only characteristics known at the beginning of pregnancy that might identify women at high risk of severe acute maternal morbidity and thus potentially improve their orientation and initial care. Second, because clinical situations may change significantly during pregnancy, we integrated the information collected over its course about potential complications that might constitute additional risk factors.

The characteristics analyzed as risk factors at the beginning of pregnancy were maternal age, maternal country of birth, prepregnancy body mass index, *preexisting* insulin-treated diabetes, preexisting hypertension, other preexisting chronic conditions, parity, previous caesarean delivery, mode of conception, and chorionicity. To identify risk factors for severe acute maternal morbidity and calculate adjusted relative risks (RR) with their 95% confidence intervals (CI), we used a multivariate Poisson regression model with a random intercept to take variability between centers into account. We then used the classification and regression tree (CART) descriptive and non-explanatory approach, [9,10] that is, we performed a CART analysis of the risk factors identified in the multivariate analysis to define and rank the factors most predictive of the risk of severe acute maternal morbidity and to individualize high-risk clinical subgroups. CART is a recursive partitioning statistical method that examines the dataset to find the best variables for grouping the women with and without severe acute maternal morbidity. Factors that are both frequent and discriminating rise in importance and result in groupings that bear resemblance and relevance to clinical practice. Among all the variables considered, CART selected the single factor that best separated the women with and without severe acute maternal morbidity to form the first node. The same procedure was then applied to each "child" node, which found the next most discriminating factor. For each node of the tree, we calculated the confidence interval of the severe acute maternal morbidity rate.

To identify the risk factors arising during pregnancy and therefore high-risk subgroups at the end of pregnancy, we repeated these statistical analyses (multivariate Poisson regression and CART analysis), adding the following variables: *gestational* insulin-treated diabetes, gestational hypertension, non-severe preeclampsia, placenta praevia, twin-to-twin transfusion syndrome, premature rupture of membranes, macrosomia for either twin (birth weight > 95$^\text{th}$

percentile of the distribution of birth weights in this cohort), and hospital characteristics, including annual volume of twin deliveries and level of care.

Two sensitivity analyses were performed. First, in order to evaluate if non-severe pre-eclampsia is a risk factor of severe acute maternal morbidity events other than severe hypertensive complications and to evaluate if it is a discriminant factor in CART analysis not only because severe preeclampsia is part of the severe acute maternal morbidity definition, we excluded cases of severe acute maternal morbidity only due to hypertensive complications. Second, to explore whether the associations found for characteristics present at the end of pregnancy may be due to differences in subsequent delivery context, we conducted an additional analysis of risk factors also including delivery-related characteristics, i.e gestational age at and mode of delivery. For this analysis, we excluded the severe acute maternal morbidity events before labor (antepartum, n = 32) or at an unknown time (n = 2), since those could not have been caused by delivery.

The proportion of women with missing data for any covariate included in the main multivariate model ranged from 0% to 11.4%. There were 7438 (84.3%) women with no missing data, 1201 (13.6%) with only one missing data item, and 184 (2.1%) with at least two missing items for covariates included in the multivariate model. Characteristics of the women with full data were similar to those with missing data (data not shown). We used multiple imputation-chained equations to impute missing data and generated 16 independent imputation data sets.

STATA 13 software (StataCorp LP, College Station, TX) was used for the descriptive and multivariate analyses. R Software Package (The R Foundation for Statistical Computing) was used for the CART analysis, in particular, the "rpart" R package.

The national data protection authority (CNIL, DR-2013-528), the consultative committee on the treatment of information on personal health data for research purposes (13–298), and the committee for the protection of people participating in biomedical research of Paris Ile-de-France 7 (PP-13-014) all approved this study. They approved that this observational study waived the need to obtain written informed consent according to the French law.

## Results

Among the 8823 women with twin pregnancies, 542 developed severe acute maternal morbidity, for a global incidence of 6.14% (95% CI, 5.64–6.64).

The main underlying causal condition of severe acute maternal morbidity was severe postpartum haemorrhage, accounting for 77.5% (n = 420, 4.76/100 twin pregnancies) of these cases (S2 Table). Admission to an ICU occurred in 22.3% (n = 121, 1.37/100 twin pregnancies). One woman in the cohort died of acute cardiac arrhythmia before labor, at 32 weeks of gestation.

Patient characteristics are presented in Table 1, with the severe acute maternal morbidity rate by maternal characteristics.

At the beginning of pregnancy, the risk factors for severe acute maternal morbidity identified in the multivariate analysis were maternal birth in sub-Saharan Africa (aRR, 1.6, 95% CI, 1.1–2.3), nulliparity (aRR, 1.6, 95% CI, 1.3–2.0), preexisting insulin-treated diabetes (aRR, 2.2, 95% CI, 1.1–4.4), and IVF with either autologous oocytes (aRR, 1.3, 95% CI, 1.0–1.6) or oocyte donation (aRR, 2.0, 95% CI, 1.4–2.8) (Table 2). Notably, maternal age, body mass index, and chorionicity were not significantly associated with the risk of severe acute maternal morbidity in women with twin pregnancies after adjustment for the other covariates. Among risk factors present at the beginning of pregnancy, CART analysis showed that oocyte donation was the most discriminating (position A, Fig 1). In the absence of oocyte donation, the severe acute maternal morbidity rate was 5.8% (95% CI, 5.3–6.3) (position B) whereas with oocyte donation, it was 14.0% (95% CI, 10.3–17.7) (position C). When factors were combined along the

**Table 1. Risk of severe acute maternal morbidity according to characteristics of the mother, pregnancy, labor, and delivery.**

| | Overall JUMODA cohort n (col %) | Women with SAMM n | Rate of SAMM per 100 twin pregnancies |
|---|---|---|---|
| Overall | 8823 (100.0) | 542 | 6.14* |
| Age (mean ± SD, years) | 31.6±5.4 | | |
| <30 | 3121 (35.4) | 184 | 5.90 |
| [30–35[ | 3225 (36.6) | 200 | 6.20 |
| [35–40[ | 1806 (20.5) | 90 | 4.98 |
| ≥40 | 671 (7.6) | 68 | 10.13 |
| Country of birth: | | | |
| Europe | 6373 (81.5) | 376 | 5.90 |
| North Africa | 848 (10.8) | 58 | 6.84 |
| Sub-Saharan Africa | 476 (6.1) | 37 | 7.77 |
| Other | 124 (1.6) | 10 | 8.06 |
| BMI before pregnancy (mean ± SD,Kg.m-2) | 24.1±5.1 | | |
| <18.5 | 562 (6.7) | 39 | 6.94 |
| [18.5–24.9] | 5054 (60.0) | 328 | 6.49 |
| [25–29.9] | 1763 (20.9) | 99 | 5.62 |
| [30–34.9] | 696 (8.3) | 34 | 4.89 |
| ≥35 | 344 (4.1) | 16 | 4.65 |
| Parity and previous caesarean | | | |
| Nulliparous | 4204 (47.8) | 323 | 7.68 |
| Parous with no previous caesarean | 3535 (40.2) | 154 | 4.36 |
| Parous with previous caesarean | 1057 (12.0) | 62 | 5.87 |
| Smoking during pregnancy | 1280 (15.0) | 55 | 4.30 |
| Preexisting insulin-treated diabetes | 53 (0.6) | 7 | 13.21 |
| Preexisting chronic hypertension | 97 (1.1) | 7 | 7.22 |
| Other preexisting chronic condition** | 665 (7.6) | 46 | 6.92 |
| Mode of conception | | | |
| Spontaneous | 5890 (67.3) | 311 | 5.28 |
| Ovulation-inducing drugs alone | 854 (9.8) | 53 | 6.21 |
| In vitro fertilization with autologous oocytes | 1675 (19.1) | 128 | 7.64 |
| Oocyte donation | 329 (3.8) | 46 | 13.98 |
| Chorionicity: | | | |
| Dichorionic | 6992 (79.7) | 445 | 6.36 |
| Monochorionic | 1781 (20.3) | 95 | 5.33 |
| Insulin-treated gestational diabetes | 325 (3.7) | 23 | 7.08 |
| Twin-to-twin transfusion syndrome | 250 (2.8) | 11 | 4.40 |
| Gestational hypertension | 502 (5.7) | 61 | 12.15 |
| Non-severe preeclampsia | 871 (9.9) | 132 | 15.15 |
| Placenta praevia | 72 (0.8) | 16 | 22.22 |
| Premature rupture of membranes | 797 (9.1) | 33 | 4.14 |
| Preterm labor | 2881 (32.8) | 167 | 5.80 |
| Gestational age at delivery (weeks days) (mean) | 35 5/6 | | |
| <32 0/7 | 895 (10.2) | 57 | 6.37 |
| 32 0/7-34 6/7 | 849 (9.7) | 43 | 5.06 |
| 35 0/7-36 6/7 | 3047 (34.6) | 178 | 5.84 |
| 37 0/7-38 6/7 | 3492 (39.7) | 230 | 6.59 |
| ≥ 39 0/7 | 512 (5.8) | 30 | 5.86 |

*(Continued)*

**Table 1.** (Continued)

| | Overall JUMODA cohort n (col %) | Women with SAMM n | Rate of SAMM per 100 twin pregnancies |
|---|---|---|---|
| Mode of delivery*** : | | | |
| Vaginal for both twins | 4216 (47.9) | 185 | 4.39 |
| Caesarean before labor | 3511 (39.9) | 257 | 7.32 |
| Caesarean during labor | 1072 (12.2) | 96 | 8.96 |
| Macrosomia for either twin**** | 575 (6.5) | 60 | 10.43 |
| Annual volume of twin deliveries: | | | |
| <50 | 2941 (33.3) | 153 | 5.20 |
| [50–99] | 2538 (28.8) | 149 | 5.87 |
| ≥100 | 3344 (37.9) | 240 | 7.18 |
| Level of care: | | | |
| I | 152 (1.7) | 4 | 2.63 |
| II | 3316 (37.6) | 168 | 5.07 |
| III | 5355 (60.7) | 370 | 6.91 |

* 95% confidence interval (CI) 5.6–6.6

** Other preexisting chronic condition, defined by a binary variable as the presence of at least one of the following: non-insulin-treated diabetes, disease of the circulatory, respiratory, or digestive system, hematological, mental, liver, or autoimmune disease, venous thromboembolism, epilepsy, nephropathy, multiple sclerosis, neoplasia, HIV infection, or active hepatitis B or C

*** Only one mode of delivery was considered for each woman—that of the second twin in case of discrepancy between the twins

**** macrosomia defined as a birth weight ≥ 95th percentile of the distribution of birth weights in the JUMODA cohort

SAMM, severe acute maternal morbidity

BMI, Body mass index

SD, standard deviation

tree, women with the highest risk of severe acute maternal morbidity were nulliparous women with either oocyte donation (14.7%; 95% CI, 10.3–19.1) (position D) or of sub-Saharan origin (12.2%, 95% CI, 6.8–17.6) (position E); these two subgroups represented 2.8% and 1.6% of the women in the JUMODA cohort, respectively.

The second multivariate model included factors identified over the course of pregnancy; placenta praevia (aRR, 3.5, 95% CI, 2.3–5.3), non-severe preeclampsia (aRR, 2.5, 95% CI, 1.9–3.2), and macrosomia for either twin (aRR, 1.7, 95% CI, 1.3–2.1) were then risk factors for severe acute maternal morbidity (Table 2). The second CART analysis, including the risk factors identified during pregnancy, showed that non-severe preeclampsia was the most discriminating factor (position A, Fig 2). In the absence of non-severe preeclampsia, the severe acute maternal morbidity rate was 5.2% (95% CI, 4.7–5.7) (position B), while with non-severe preeclampsia, it reached 15.2% (95% CI, 12.8–17.6) (position C). As we followed the "non-severe preeclampsia" branch to the terminal leaves of the tree, the highest risk of severe acute maternal morbidity was found in two subgroups of women with non-severe preeclampsia: those with oocyte donation (28.9%; 95% CI, 19.9–37.9) (position D) and those nulliparas born in sub-Saharan Africa (26.9%; 95% CI, 9.9–43.9) (position E); these subgroups accounted respectively for 1.1% and 0.3% of the women in the JUMODA cohort. Conversely, the women at lowest risk (3.8%; 95% CI, 3.2–4.4) were multiparous with none of the following events: non-severe preeclampsia, placenta praevia, or macrosomia (position F).

In the sensitivity analysis excluding severe acute maternal morbidity related to hypertensive complications, results were similar and notably non-severe preeclampsia remained a risk factor of severe acute maternal morbidity with a similar estimate for adjusted relative risk (S3

**Table 2. Risk factors for severe acute maternal morbidity in twin pregnancies, JUMODA cohort (N = 8823).**

| Potential risk factors | | Crude RR | At the beginning of pregnancy Adjusted RR* | At the end of pregnancy Adjusted RR* |
|---|---|---|---|---|
| | | (95% CI) | (95% CI) | (95% CI) |
| Maternal age (years) | | | | |
| | <30 | Reference | Reference | Reference |
| | [30–35[ | 1.1 (0.9–1.2) | 1.1 (0.9–1.3) | 1.0 (0.9–1.3) |
| | [35–40[ | 0.9 (0.7–1.1) | 0.8 (0.7–1.1) | 0.8 (0.6–1.0) |
| | ≥40 | 1.7 (1.3–2.2) | 1.3 (0.9–1.8) | 1.2 (0.9–1.7) |
| Body mass index before pregnancy (Kg.m-2) | | | | |
| | <18,5 | 1.1 (0.8–1.5) | 1.1 (0.8–1.5) | 1.1 (0.8–1.6) |
| | [18.5–24.9] | Reference | Reference | Reference |
| | [25–29.9] | 0.9 (0.7–1.1) | 0.9 (0.7–1.1) | 0.8 (0.7–1.0) |
| | [30–34.9] | 0.8 (0.5–1.1) | 0.8 (0.5–1.1) | 0.7 (0.5–1.0) |
| | ≥35 | 0.7 (0.6–1.0) | 0.8 (0.5–1.4) | 0.7 (0.4–1.3) |
| Country of birth | | | | |
| | Europe | Reference | Reference | Reference |
| | North Africa | 1.2 (0.9–1.5) | 1.3 (0.9–1.7) | 1.3 (0.9–1.6) |
| | Sub-Saharan Africa | 1.3 (1.0–1.8) | 1.6 (1.1–2.3) | 1.4 (1.0–2.1) |
| | Other | 1.4 (0.8–2.5) | 1.3 (0.6–2.8) | 1.4 (0.7–2.9) |
| Parity and previous caesarean | | | | |
| | Nulliparous | 1.8 (1.5–2.1) | 1.6 (1.3–2.0) | 1.5 (1.2–1.9) |
| | Parous without previous caesarean | Reference | Reference | Reference |
| | Parous with previous caesarean | 1.4 (1.0–1.8) | 1.3 (0.9–1.8) | 1.3 (0.9–1.8) |
| Preexisting hypertension | | 1.2 (0.6–2.4) | 1.1 (0.6–2.3) | 0.8 (0.4–1.6) |
| Preexisting insulin-treated diabetes | | 2.2 (1.1–4.3) | 2.2 (1.1–4.4) | 1.6 (0.8–3.0) |
| Other preexisting chronic condition** | | 1.1 (0.9–1.5) | 1.0 (0.8–1.4) | 1.1 (0.8–1.5) |
| Mode of conception | | | | |
| | Spontaneous | Reference | Reference | Reference |
| | Ovulation-inducting drugs alone | 1.2 (0.9–1.6) | 1.1 (0.8–1.4) | 1.1 (0.8–1.4) |
| | In vitro fertilization with autologous oocytes | 1.5 (1.2–1.8) | 1.3 (1.0–1.6) | 1.3 (1.0–1.6) |
| | Oocyte donation | 2.7 (2.0–3.5) | 2.0 (1.4–2.8) | 1.7 (1.2–2.3) |
| Chorionicity | | | | |
| | Dichorionic | Reference | Reference | Reference |
| | Monochorionic | 0.8 (0. 7–1.0) | 0.9 (0.8–1.2) | 1.0 (0.8–1.2) |
| Insulin-treated gestational diabetes | | 1.2 (0.8–1.7) | - | 1.0 (0.7–1.6) |
| Gestational hypertension | | 2.1 (1.6–2.7) | - | 1.2 (0.9–1.7) |
| Non-severe preeclampsia | | 2.9 (2.4–3.5) | - | 2.5 (1.9–3.2) |
| Placenta praevia | | 3.7 (2.4–5.7) | - | 3.5 (2.3–5.3) |
| Twin-to-twin transfusion syndrome | | 0.7 (0.4–1.3) | - | 0.9 (0.5–1.7) |
| Premature rupture of membranes | | 0.7 (0.5–0.9) | - | 0.7 (0.5–1.0) |
| Macrosomia*** | | 1.5 (1.2–1.8) | - | 1.7 (1.3–2.1) |
| Maternity hospital: | | | | |
| Annual volume of twin deliveries | | | | |
| | <50 | 0.7 (0.6–0.9) | - | 1.1 (0.7–1.6) |
| | [50–99] | 0.8 (0.7–1.0) | - | 1.0 (0.8–1.2) |
| | ≥100 | Reference | - | Reference |
| Level of care | | | | |
| | I | 0.4 (0.1–1.0) | - | 0.4 (0.1–1.4) |
| | II | 0.7 (0.6–0.9) | - | 0.7 (0.5–1.0) |

(*Continued*)

**Table 2.** (Continued)

| Potential risk factors | | Crude RR | At the beginning of pregnancy | At the end of pregnancy |
|---|---|---|---|---|
| | | | Adjusted RR* | Adjusted RR* |
| | | (95% CI) | (95% CI) | (95% CI) |
| | III | Reference | - | Reference |

RR, relative risk; CI, confidence interval

*Each relative risk is adjusted for all other variables in the column, multilevel multivariate Poisson regression model, with imputed data

** Other preexisting chronic condition, defined by a binary variable as the presence of at least one of the following: non-insulin-treated diabetes, disease of the circulatory, respiratory, or digestive system, hematological, mental, liver, or autoimmune disease, venous thromboembolism, epilepsy, nephropathy, multiple sclerosis, neoplasia, HIV infection, or active hepatitis B or C

*** macrosomia defined as a birth weight ≥ 95th percentile of the distribution of birth weights in the JUMODA cohort

Table). In addition, CART analysis showed that non-severe preeclampsia remained the most discriminating factor for identifying women at higher risk of severe acute maternal morbidity (S1 Fig).

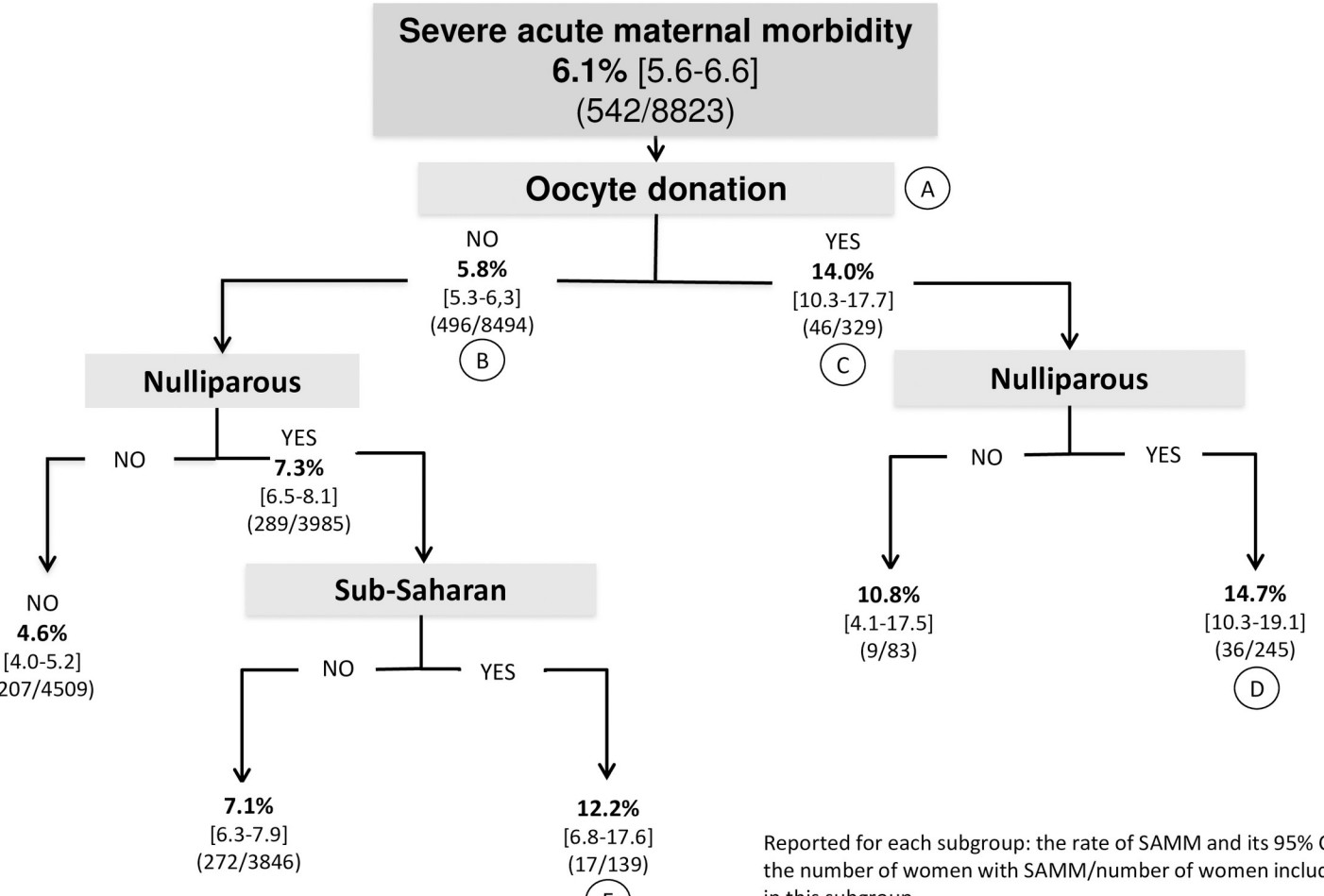

**Fig 1. Factors present at the beginning of pregnancy.** Classification and regression tree analysis: hierarchy of factors associated with severe acute maternal morbidity, number of women, and percentage of events at each node Reported for each subgroup: the rate of SAMM and its 95% CI, the number of women with SAMM/number of women included in this subgroup.

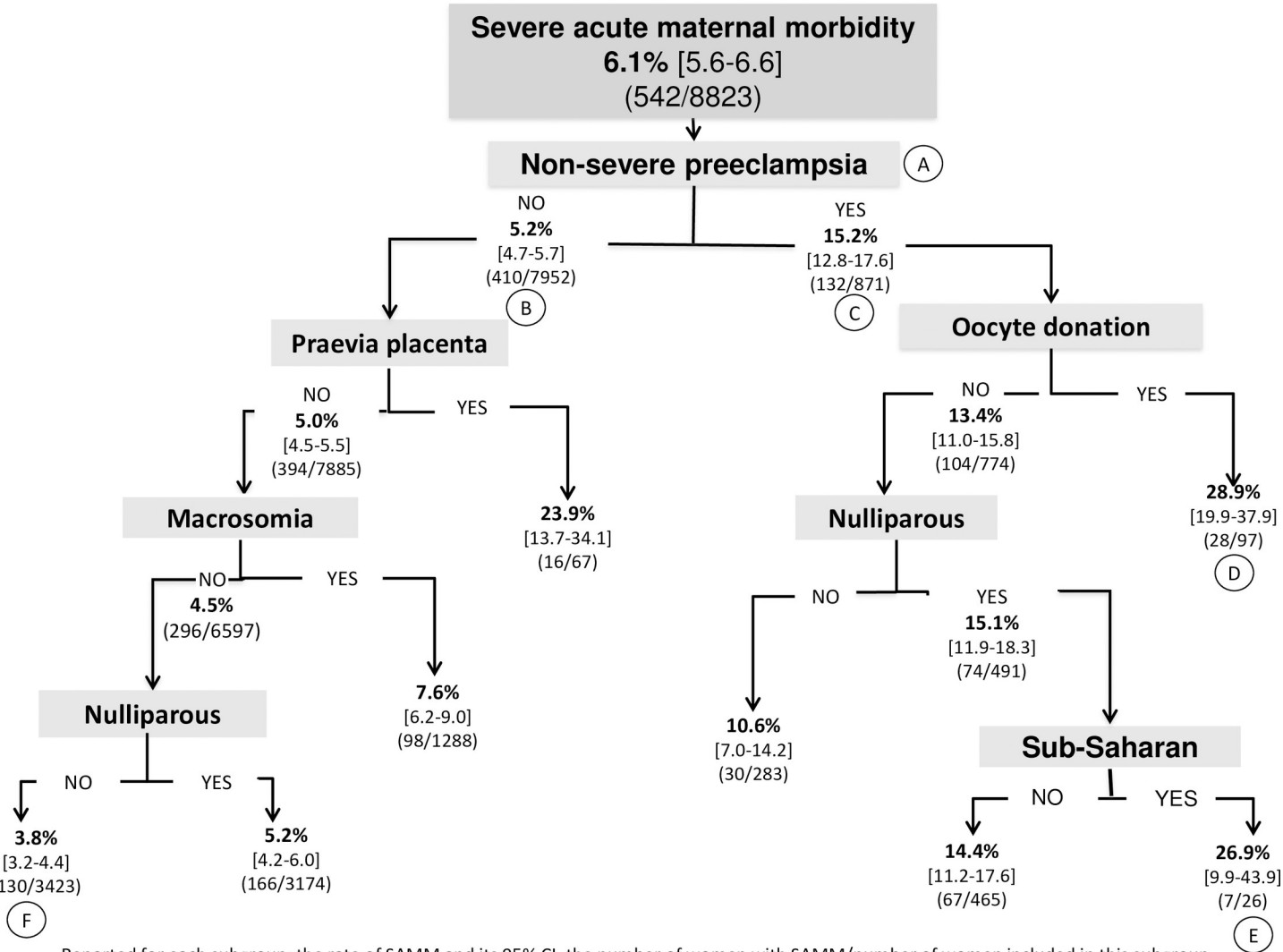

Reported for each subgroup: the rate of SAMM and its 95% CI, the number of women with SAMM/number of women included in this subgroup

**Fig 2. Factors present at the beginning and arising during pregnancy.** Classification and regression tree analysis: hierarchy of factors associated with severe acute maternal morbidity, number of women, and percentage of events at each node Reported for each subgroup: the rate of SAMM and its 95% CI, the number of women with SAMM/number of women included in this subgroup.

The sensitivity analysis including delivery-related characteristics identified the same risk factors as above. In addition, caesarean delivery was associated with a significantly higher risk of intra- or postpartum severe acute maternal morbidity, whether performed before (aRR, 1.3, 95% CI, 1.0–1.6) or during labor (aRR, 1.6, 95% CI, 1.2–1.9). Finally a gestational age at birth less than 37 weeks of gestation was associated with a significantly lower risk of intra- or postpartum severe acute maternal morbidity (aRR, 0.8, 95% CI, 0.7–0.9) (S4 Table).

## Discussion

### Main findings

In women with twin pregnancies, the overall increased risk of developing severe acute maternal complications varies by subgroups of women. At the beginning of twin pregnancy, nulliparous women with oocyte donation were identified as those with the highest risk of severe acute

maternal morbidity. At the end of pregnancy, two subgroups of women with a risk exceeding 25% of developing severe acute maternal morbidity were identified: those with both oocyte donation and non-severe preeclampsia, and nulliparas born in sub-Saharan Africa with non-severe preeclampsia.

## Strengths and limitations

Our study has several strengths. It was a large population-based study including a substantial number of both twin pregnancies and of cases of severe acute maternal morbidity. The analysis of severe acute maternal morbidity was planned during the design of the JUMODA study, so that the data to characterize it were defined in advance and collected prospectively. The data collection method involving a manual review of the medical records of each woman provided detailed and accurate information on maternal and pregnancy characteristics, unlike data extracted from routine hospital databases in retrospective studies.

One limitation of this study is that the JUMODA cohort included only maternity units with more than 1500 annual deliveries. Although the twin deliveries included in the JUMODA study accounted for 75% of all twin deliveries in France over the study period, this could potentially mean that our results cannot be fully generalized to women giving birth in the smallest hospitals. However, the incidence of severe acute maternal morbidity we found here is similar to the one reported in another French population-based study including units of all sizes.[3] Furthermore, despite more than 8800 women were included in this study, its statistical power for covering rare pathologies that might be severe acute maternal morbidity risk factors remains limited. Finally, the results of CART analyses would benefit from a validation in another cohort of twin pregnancies and are limited by the fact that not all clinical situations are represented in the final tree.

## Interpretation

The most original result—and perhaps the most helpful for clinicians—is the characterization among this population of subgroups with various levels of risk of severe acute maternal morbidity, by CART analysis. This analysis is complementary to the standard epidemiologic study of risk factors by multivariate regression, but applies an approach that is more pragmatic than explanatory. By directly estimating the severe acute maternal morbidity rate in subgroups that combine multiple risk factors, it may help clinicians to advise women on the most appropriate place for delivery and to adapt their management throughout pregnancy by anticipating the occurrence of those adverse events.

At the beginning of pregnancy, we found that maternal birth in sub-Saharan Africa, preexisting insulin-treated diabetes, nulliparity and mode of conception were risk factors for severe acute maternal morbidity. Although these results are concordant with previous studies conducted in general populations of parturients or populations of singleton pregnancies, it is important to verify that these risk factors persist in this high-risk population [11–17]. CART analysis found that at the beginning of pregnancy oocyte donation is the most discriminating factor for severe acute maternal morbidity. This increased maternal risk in women with twin pregnancies after IVF contradicts some previous studies exploring this association [9,18–20]. The latter, however, were limited by their inability to differentiate between spontaneous pregnancy and ovulation-inducing drugs alone and between in vitro fertilization with autologous and donated oocytes.

An unexpected result was the lack of association between severe acute maternal morbidity and maternal age. Wondering whether this result might be due to overadjustment when mode of conception and age were simultaneously included in the model, we conducted an analysis

stratified by mode of conception. It confirmed the lack of association, regardless of the mode of conception (data not shown).

At the end of pregnancy, the CART analysis highlighted that even in subgroups of twin pregnancies with the lowest risk, the rate of severe acute maternal morbidity still remains twice the rate usually reported in singleton pregnancies [3,9–12]. We found that the most discriminating factor for the occurrence of severe acute maternal morbidity was non-severe preeclampsia, which has been previously reported as a risk factor for severe acute maternal morbidity in general populations of parturients and in singleton pregnancies [21,22]. We confirm here that this association also exists among twin pregnancies. This result might appear trivial, since non-severe preeclampsia could be considered just a step within the morbidity continuum of hypertensive-related complications. Nonetheless, it is noteworthy that when we excluded severe acute maternal morbidity events due to hypertensive complications from the outcomes, results of multivariate and CART analyses remained similar. This suggests that non-severe preeclampsia is a risk factor for other causes of severe acute maternal morbidity than hypertensive complications. Moreover, the CART analysis provided here shows that among women with twin pregnancies and non-severe preeclampsia, additional risk factors boost the risk of a severe acute maternal morbidity up to rates above 25%.

The identification of some modifiable risk factors can help improve the management of women with twin pregnancy. Preventive care can start during the preconceptional period. The high maternal risk in twin pregnancies after in vitro fertilization constitutes an argument for limiting the number of embryos transferred to a single embryo in order to prevent medically-induced multiple pregnancies and associated severe maternal morbidity. This is particularly important in case of oocyte donation, which must be a reasoned practice. An increased risk of severe acute maternal morbidity associated with maternal country of birth, another identified risk factor, may also be indirectly modifiable. Some previous studies have reported an increased risk of severe hypertensive complications in migrant women, in particular from sub-Saharan Africa [11,23,24]. Among suggested hypotheses are genetic factors [25] but also differential prenatal care [26, 27,28]. Thus if the association between maternal country of birth and severe acute maternal morbidity reflects differential prenatal care, this risk factor could be changed by improving the quality of care for these vulnerable women.

## Conclusion

About one in 17 women with a twin pregnancy will develop severe acute maternal morbidity overall, and this proportion rises up to more than a quarter in some particular subgroups of women. These results have implications for clinical practice. They will help identifying modifiable risk factors, personalizing information and improving shared decision regarding prenatal and delivery care for women with twin pregnancies, according to their individual profile.

## Supporting information

**S1 Table. EPIMOMS Multicriteria Standardized Definition of Severe Acute Maternal Morbidity, Developed Through a National Delphi Formal Expert Consensus Process.**
(DOC)

**S2 Table. Incidence, timing, and underlying causal conditions of severe acute maternal morbidity in twin pregnancies in the JUMODA cohort.** * pregnancy: delivered at or after 22 weeks of gestation
** 95% confidence interval (CI) 5.6–6.6

*** nonexclusive categories.
(DOC)

**S3 Table. Risk factors for severe acute maternal morbidity in twin pregnancies, sensitivity analysis after exclusion of women with severe acute maternal morbidity only due to hypertensive complication (19 cases), JUMODA cohort.** (n = 8804 women).
RR, relative risk; CI, confidence interval
*Each relative risk is adjusted for all other variables in the table, multilevel multivariate Poisson regression model, with imputed data.
(DOC)

**S4 Table. Risk factors for intrapartum and postpartum severe acute maternal morbidity in twin pregnancies, sensitivity analysis including delivery-related characteristics, JUMODA cohort (n = 8789 women).** RR, relative risk; CI, confidence interval
*Each relative risk is adjusted for all other variables in the table, multilevel multivariate Poisson regression model.
(DOC)

**S1 Fig. Classification and regression tree analysis: Factors present at the beginning and arising during pregnancy, sensitivity analysis after exclusion of women with severe acute maternal morbidity only due to hypertensive complication (19 cases), JUMODA cohort (n = 8804 women).**
(PPT)

## Acknowledgments

The authors thank URC-CIC Paris Descartes Necker/Cochin (Laurence Lecomte) for the study implementation, monitoring, and data management,all collaborators of the JUMODA (JUmeaux Mode d'Accouchement) study group: Pr Langer (CHU Hautepierre), Dr Sananes, Dr Favre (CMCO de Schiltigheim), Dr Kutnahorsky (CMC de Colmar), Mme Fessler (CHR de Mulhouse), Dr Lehmann (CHR d'Haguenau), Dr Adame (Clinique Sainte-Anne, Strasbourg), Dr Plemere (Clinique Sainte-Anne, Strasbourg), Dr Chabanier (CHU de Bordeaux), Dr Trebesses (Clinique Bagatelle, Talence), Dr Poumier-Chabannier (CH de Bayonne), Dr Defert (CH de Mont de Marsan), Dr Bohec (CH de Pau), Dr Collin (Polyclinique de Navarre, Pau), Dr Venditelli (CHU de Clermont-Ferrand), Dr Deffarges, Dr Vidal (Clinique de la Chataigneraie, Beaumont), Dr Desvignes (CH de Vichy), Pr Dreyfus (CHU de Caen), Dr Samuel (CH du Puy-en-Velay), Dr Beucher (CHU de Caen), Dr Dolley (CHU de Caen), Dr Durin (Clinique du Parc, Caen), Dr Six (CH d'Avranches), Dr Beniada (CH de Lisieux), Dr Balouet (CH de Saint-Lô), Dr Desprès (CH de Cherbourg), Mme Mathis (CH de Cherbourg), Pr Sagot (CHU de Dijon), Dr Yacoub (CHU de Dijon), Dr Bulot (CH de Chalon-sur-Saône), Dr Dellinger (CH d'Auxerre), Dr Spagnolo (CH de Mâcon), Pr Poulain (CHU de Rennes), Dr Moquet (Clinique de la Sagesse, Rennes), Mme Bourgault (Clinique de la Sagesse, Rennes), Dr Seconda (CHP Saint-Grégoire), Dr Moinon (CH de Saint-Brieuc), Dr Roy-Dahhou (CH de Saint-Malo), Dr Pittion (CH Bretagne Sud, Lorient), Dr Chauveau (CH Bretagne Atlantique, Vannes), Dr Laurent (CHU de Brest), Dr Lelièvre (CHU de Brest), Dr Bellot (CH de Quimper), Dr Salnelle (Polyclinique de Keraudren, Brest), Pr Perrotin (CHRU de Tours), Dr Ramos (CH d'Orléans), Dr Montmasson (CH de Blois), Dr Ollivier (CH de Chartres), Dr Hoock (CH de l'agglomération montargoise), Dr Ben Romdhane (CH de l'agglomération montargoise), Pr Graesslin (CHU de Reims), Dr Méreb (CH de Charleville Mézières), Pr Riethmuller (CHU de Besançon), Dr Boyadjian (CH de Pontarlier), Dr Gannard (CH de Dole), Dr Levy (CH de Belfort), Dr Reviron (CH de Lons le

Saunier), Pr Verspyck, Pr Marpeau (CHU de Rouen), Dr Durand Reville (Clinique Mathilde, Rouen), Dr Talbot (CH Le Havre), Dr Mathieu (CH d'Elbeuf), Dr Machevin (CH d'Evreux), Dr Truong Canh (CH de Vernon), Dr Guillon (CH du Belvédère, Mont Saint-Aignan), Dr Ménard (CHU Cochin-Port Royal), Dr Bourgeois Moine (CHU Bichat), Pr Nizard (CHU Pitié Salpêtrière), Pr Dommergues (CHU Pitié Salpêtrière), Dr De Carné Carnavalet (CHU Trousseau), Dr Lemercier (CHU Necker Enfants Malades), Dr Bornes (CHU Tenon), Dr Ricbourg (CHU Lariboisière), Dr Harvey (Hôpital des Diaconesses), Dr Azarian (Institut Mutualiste Montsouris), Dr Azria (Groupe Hospitalier Saint Josep), Pr Kayem (CHU Louis Mourier), Pr Benachi (CHU Antoine Béclère), Dr Ceccaldi (CHU Beaujon), Pr Sénat (CHU Bicêtre), Dr Galimard (CH de Neuilly), Dr Picone (Hôpital Foch), Dr Bounan (CH de Saint-Denis), Dr Hatem (CH de Saint-Denis), Pr Poncelet (CH de Montreuil), Pr Carbillon (CHU Jean Verdier), Pr Haddad (CHI de Créteil), Dr Pachy (Hôpitaux de Saint Maurice Esquirol), Mme Deshons (CH de Pontoise), Dr Colliaut Espagne (CH de Montmorency), Pr Rozenberg (CHI de Poissy), Dr Raynal (CH de Versailles), Dr Godard (CH de Mantes la Jolie), Dr Soltane (CH de Villeneuve Saint-Georges), Dr Piel (CH de Villeneuve Saint-Georges), Dr Abbara (CH de Longjumeau), Dr Rigonnot (CH du Sud Francilien, Corbeil Essonne), Dr Jault (CH de Melun), Dr Marchaudon (CH de Fontainebleau), Dr Moumen (CH de Meaux), Dr Wafo (CH de Lagny), Pr De Tayrac (CHU de Nîmes), Dr Léonard (Polyclinique Grand Sud, Nîmes), Dr Terschiphorst (Polyclinique Kennedy, Nîmes), Dr Vintejoux (CHU de Montpellier), Dr Filippi (Clinique Clémentville, Montpellier), Dr Rouard (Clinique Saint-Roch, Montpellier), Dr Galtier (CH de Béziers), Dr Cogan (CH de Carcassonne), Dr Koninck (CH de Perpignan), Pr Morel (CHU de Nancy), Dr Dahlhoff Rodriguez (CH de Metz), Dr Collin (CH de Thionville), Pr Parant (CHU de Toulouse), Dr Thévenot (Clinique Sarrus), Dr Cére (Clinique Sarrus), Pr Deruelle (CHRU de Lille), Dr Clouqueur (CHRU de Lille), Dr Pouilly (Polyclinique du Bois, Lille), Dr Denoit (GHIC Saint-Vincent-de-Paul, Lille), Dr Régis (CH d'Armentières), Dr Rivaux (CH d'Armentières), Dr Legoueff (CH de Roubaix), Dr Jambon (CH de Tourcoing), Dr Bory (CH de Seclin), Dr Sendon (CH de Valenciennes), Dr Tillouche (CH de Valenciennes), Dr Boodhun (CH de Dunkerque), Dr Bothuyne (CH de Lens), Dr Sicot (CH de Boulogne-sur-Mer), Dr Brochot (CH d'Arras), Dr Carillon (CH de Calais), Dr Coudoux (CH de Calais), Dr Notteau (CH de Saint-Omer), Dr Hautemonte (CHU Marseille, Hôpital Nord), Pr D'Ercole (CHU Marseille, Hôpital Nord), Dr Heckenroth (CHU Marseille, Hôpital La Conception), Dr Desbrière (CH Saint-Joseph), Dr Volle (CH de Martigues), Dr Mauviel (CH de Toulon), Dr Danoy (CH d'Aix-en-Provence), Dr Marpeau (Clinique l'Etoile-Maternité catholique de Provence, Aix en Provence), Dr David (CH de Salon de Provence), Dr Lepreux (CH d'Avignon), Dr Leroux Hilmi (CHU de Nice), Dr Adrados (CHU de Nice), Pr Bongain (CHU de Nice), Mme Roulant (Clinique Saint-Georges, Nice), Dr Kaemmerlen (CH de Grasse), Dr Duforestel (CH d'Antibes), Dr De Jesus (CH de Cannes), Pr Winer (CHU de Nantes), Dr Paumier (Polyclinique de l'Atlantique, Nantes), Dr Lebret-Colau (Clinique Jules Verne, Nantes), Dr Troche (CH de Saint-Nazaire), Pr Sentilhes (CHU d'Angers), Dr Chève (CH Le Mans), Dr Moya (CH de Saumur), Dr Karirisi (CH de Laval), Dr Pasco (CH de Cholet), Dr Ducarme (CH de La Roche-sur-Yon), Pr Gondry (CHU d'Amiens), Dr Théret (CHU d'Amiens), Mme Buisson (Groupe Santé Victor Pauché, Amiens), Dr Urbaniack (CH de Beauvais), Dr Dienga (CH de Creil), Dr Closset (CH de Saint-Quentin), Dr Touzart (CH de Compiègne), Pr Pierre (CHU de Poitiers), Dr Leborgne (CH de La Rochelle), Dr Lathélize (CH de Rochefort), Dr Chauvet (CH de Niort), Dr Sarreau (CH d'Angoulême), Dr Bretheau (CH de Saintes), Dr Godard (CH de Châtellerault), Dr Yannoulopoulos (CH Nord Deux Sèvres, Bressuire), Pr Aubard (CHU de Limoges), Pr Rudigoz (CHU La Croix Rousse, Lyon), Mme Dupont (CHU La Croix Rousse, Lyon), Pr Dupuis (CHU Lyon Sud, Lyon), Dr Battie (CHU Mère-Enfant, Lyon), Dr Mein (Hôpital Natecia, Lyon), Dr Mossan-Lourcy (Clinique du Val d'Ouest, Ecully), Dr Rane (Clinique Saint-Vincent-de-Paul, Bourgoin-Jallieu), Dr Fernandez (CH de

Valence), Dr Sayegh (CH de Villefranche), Dr Dirix (CH de Montélimar), Dr Nord (CH de Roanne), Pr Chauleur (CHU de Saint-Etienne), Dr Hugot (CH de Bourg-en-Bresse), Dr Ferlay (CH de Bourg-en-Bresse), Dr Equy (CHU de Grenoble), Dr Canonica (Clinique Belledonne, Grenoble), Dr Gaillard (CH de Voiron), Dr Dubois (CH de Chambéry), Dr Dujardin (CH de Sallanches), Dr Braig (CH d'Annecy), Dr Deramecourt (CH d'Annemasse), Dr Vincent-Génod (CH de Thonon) ;

And all collaborators of the Groupe de Recherche en Obstétrique et Gynécologie (GROG): Thomas Schmitz (Department of Gynecology-Obstetrics, Assistance Publique–Hôpitaux de Paris, Robert Debré), Elie Azria (Department of Gynecology-Obstetrics, Groupe Hospitalier Saint Joseph), Céline Chauleur (Department of Gynecology-Obstetrics, Centre Hospitalier Universitaire de Saint Etienne), Catherine Deneux-Tharaux (UMR1153–Obstetrical, Perinatal and Paediatric Epidemiology (EPOPée Research Team), Descartes University–INSERM, Paris), Muriel Doret (Department of Gynecology-Obstetrics, Hôpital Femme Mère enfant Lyon), Anne Ego (Université Grenoble Alpes/CNRS/TIMC-IMAG UMR 5525 (Equipe ThE-MAS), Grenoble), Denis Gallot (Department of Gynecology-Obstetrics, Centre Hospitalier Universitaire, Clermont-Ferrand), François Goffinet (Department of Gynecology-Obstetrics, Assistance Publique–Hôpitaux de Paris, Cochin–Port Royal), Gilles Kayem (Department of Gynecology-Obstetrics, Assistance Publique–Hôpitaux de Paris, Trousseau), Bruno Langer (Department of Gynecology-Obstetrics Hautepierre, Hôpitaux universitaire de Strasbourg), Camille Leray (Department of Gynecology-Obstetrics, Assistance Publique–Hôpitaux de Paris, Cochin–Port Royal), Laurent Mandelbrot (Department of Gynecology-Obstetrics, Assistance Publique–Hôpitaux de Paris, Louis Mourier, Colombes), Olivier Morel (Department of Gynecology-Obstetrics, Centre Hospitalier Universitaire de Nancy), Frank Perrotin (Department of Gynecology-Obstetrics, Centre Hospitalier Universitaire de Tours), Patrick Rozenberg (Department of Gynecology-Obstetrics, Centre Hospitalier Universitaire de Poissy), Damien Subtil (Department of Gynecology-Obstetrics, Centre Hospitalier Universitaire de Lille), Christophe Vayssiere (Department of Gynecology-Obstetrics, Centre Hospitalier Universitaire de Toulouse), Norbert Winer (Department of Gynecology-Obstetrics, Centre Hospitalier Universitaire de Nantes).

## Author Contributions

**Conceptualization:** Diane Korb, Thomas Schmitz, Catherine Deneux-Tharaux.

**Formal analysis:** Diane Korb, Aurélien Seco, Catherine Deneux-Tharaux.

**Funding acquisition:** Thomas Schmitz.

**Investigation:** François Goffinet.

**Methodology:** Diane Korb, Thomas Schmitz, Aurélien Seco.

**Project administration:** Thomas Schmitz.

**Supervision:** Thomas Schmitz, Catherine Deneux-Tharaux.

**Validation:** Thomas Schmitz, François Goffinet, Catherine Deneux-Tharaux.

**Writing – original draft:** Diane Korb, Thomas Schmitz, Catherine Deneux-Tharaux.

**Writing – review & editing:** Aurélien Seco, François Goffinet.

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
