## [Decision Letter · Decision Letter 0]

3 Dec 2019

PONE-D-19-27799

Risk factors and high-risk subgroups of severe acute maternal morbidity in twin pregnancy: a population-based study

PLOS ONE

Dear Dr Korb,

Thank you for submitting your manuscript to PLOS ONE. After careful consideration, we feel that it has merit but does not fully meet PLOS ONE’s publication criteria as it currently stands. Therefore, we invite you to submit a revised version of the manuscript that addresses the points raised during the review process.

In addition to the few issues raised by the reviewers, please use SI unit (convention units may be use in parenthesis). Also use English phrase for conditions  such as “Placenta previa”.

We would appreciate receiving your revised manuscript by Jan 17 2020 11:59PM. To enhance the reproducibility of your results, we recommend that if applicable you deposit your laboratory protocols in protocols.io, where a protocol can be assigned its own identifier (DOI) such that it can be cited independently in the future. For instructions see: http://journals.plos.org/plosone/s/submission-guidelines#loc-laboratory-protocols

We look forward to receiving your revised manuscript.

Kind regards,

Pal Bela Szecsi, M.D. D.M.Sci.

Academic Editor

PLOS ONE

Journal Requirements:

Supported by a grant from the French Ministry of Health (Programme Hospitalier de Recherche Clinique, AOM2012)

The authors received no specific funding for this work.

3. Please provide additional details regarding participant consent. In the ethics statement in the Methods and online submission information, please ensure that you have specified (1) whether consent was informed and (2) what type you obtained (for instance, written or verbal). If your study included minors, state whether you obtained consent from parents or guardians. If the need for consent was waived by the ethics committee, please include this information.

7. One of the noted authors is a group or consortium: JUmeaux MODe d’Accouchement (JUMODA) study group and the Groupe de Recherche en Obstétrique et Gynécologie (GROG)

In addition to naming the author group, please list the individual authors and affiliations within this group in the acknowledgments section of your manuscript. Please also indicate clearly a lead author for this group along with a contact email address.

Reviewers' comments:

Reviewer's Responses to Questions

**Comments to the Author**

1. Is the manuscript technically sound, and do the data support the conclusions?

Reviewer #1: Yes

Reviewer #2: Partly

2. Has the statistical analysis been performed appropriately and rigorously? 

Reviewer #1: I Don't Know

Reviewer #2: Yes

3. Have the authors made all data underlying the findings in their manuscript fully available?

Reviewer #1: No

Reviewer #2: Yes

4. Is the manuscript presented in an intelligible fashion and written in standard English?

Reviewer #1: Yes

Reviewer #2: Yes

5. Review Comments to the Author

Reviewer #1: This manuscript uses a large population-based study of twin pregnancies "Jumonda" to identify risk factors for severe maternal morbidity and was planned prospectively. The primary outcome was a composite of including maternal death, severe hemorrhage (defined by need for second line therapy), transfusion >=units of pRBCs, uterine artery embolization, and several other severe adverse outcomes. They identified two subgroups of women with a >25% risk of developing severe acute maternal morbidity: those with both oocyte donation, non-severe preeclampsia, and nulliparas born in Sub-Saharan Africa with non-severe preeclampsia.

Reviewer #2: This is a potential wonderful paper. I am afraid I have major problems with the definition of severe morbidity.

Pre ecclampsia requiring delivery is certainly serious but NOT always a severe morbid event.

In a large network Criteria to admit to an ICU will not all be the same and thus potential for bias.

This has the potential for a significant impact please try to find another channel is Tiffany and why except it if I am open even though yours was conducted using a a delfi

Can you be more strict in your defn of SMM and redo ? It will be more meaningful for everyone

6. PLOS authors have the option to publish the peer review history of their article (what does this mean?). If published, this will include your full peer review and any attached files.

Reviewer #1: No

Reviewer #2: No

---

## [Author Response · Author response to Decision Letter 0]

7 Jan 2020

Dr Diane Korb

INSERM U1153, Obstetric, Perinatal, and Pediatric Epidemiology Research Team

53 avenue de l’Observatoire, 75014 Paris, France

Tel: +33(0)1 42 34 55 80 Fax: +33 (0)1 43 26 89 79

Email: diane.korb@inserm.fr

 Paris, December 20th 2019

Dear Editor, 

Thank you for your response on December 3th 2019, concerning our manuscript PONE-D-19-27799 entitled “Risk factors and high-risk subgroups of severe acute maternal morbidity in twin pregnancy: a population-based study” informing us you would be willing to give further consideration to a revised version. 

The authors are very grateful to the Reviewers and Editors for their constructive help. We think the paper has been much improved. Our revised version has taken into account all the following points raised by the Reviewers and Editors. 

All the authors have read and approved the revised version of the paper.

We hope our manuscript now meets the standards of Plos One.

Yours sincerely,

Diane Korb

All line numbers refer to the revised version of the manuscript.

Response to the Editor

In addition to the few issues raised by the reviewers, please use SI unit (convention units may be use in parenthesis). Also use English phrase for conditions such as “Placenta previa”.

As requested, we replaced American by English language.

1. Thank you for stating the following in the Acknowledgments Section of your manuscript:

Supported by a grant from the French Ministry of Health (Programme Hospitalier de Recherche Clinique, AOM2012)

The authors received no specific funding for this work.

We modified the online submission form accordingly.

2. Please provide additional details regarding participant consent. In the ethics statement in the Methods and online submission information, please ensure that you have specified (1) whether consent was informed and (2) what type you obtained (for instance, written or verbal). If your study included minors, state whether you obtained consent from parents or guardians. If the need for consent was waived by the ethics committee, please include this information.

We precised in line 161: “They approved that this observational study waived the need to obtain written informed consent according to the French law.”

3. PLOS requires an ORCID iD for the corresponding author in Editorial Manager on papers submitted after December 6th, 2016. Please ensure that you have an ORCID iD and that it is validated in Editorial Manager. To do this, go to ‘Update my Information’ (in the upper left-hand corner of the main menu), and click on the Fetch/Validate link next to the ORCID field. This will take you to the ORCID site and allow you to create a new iD or authenticate a pre-existing iD in Editorial Manager. Please see the following video for instructions on linking an ORCID iD to your Editorial Manager account: 

I added my ORCID iD in the online submission form (https://orcid.org/0000-0002-3074-8269).

4. In your Data Availability statement, you have not specified where the minimal data set underlying the results described in your manuscript can be found. PLOS defines a study's minimal data set as the underlying data used to reach the conclusions drawn in the manuscript and any additional data required to replicate the reported study findings in their entirety. All PLOS journals require that the minimal data set be made fully available. For more information about our data policy, please see https://clicktime.symantec.com/3YZSF9bEtVTXuArrvN4RTeq6H2?u=http%3A%2F%2Fjournals.plos.org%2Fplosone%2Fs%2Fdata-availability.

Upon re-submitting your revised manuscript, please upload your study’s minimal underlying data set as either Supporting Information files or to a stable, public repository and include the relevant URLs, DOIs, or accession numbers within your revised cover letter. For a list of acceptable repositories, please see https://clicktime.symantec.com/34uYZEGrViwwJiXZ15MZ2ch6H2?u=http%3A%2F%2Fjournals.plos.org%2Fplosone%2Fs%2Fdata-availability%23loc-recommended-repositories. Any potentially identifying patient information must be fully anonymized.

Important: If there are ethical or legal restrictions to sharing your data publicly, please explain these restrictions in detail. Please see our guidelines for more information on what we consider unacceptable restrictions to publicly sharing data: https://clicktime.symantec.com/3VUWv2qA9KnXzUZXsdCzx4V6H2?u=http%3A%2F%2Fjournals.plos.org%2Fplosone%2Fs%2Fdata-availability%23loc-unacceptable-data-access-restrictions. Note that it is not acceptable for the authors to be the sole named individuals responsible for ensuring data access.

The data underlying the findings cannot be made freely available because of ethical and legal restrictions. This is because the present study includes an important number of variables that, together, could be used to re-identify the participants based on a few key characteristics and then be used to have access to other personal data. Therefore, the French National Data Safety Authority (CNIL) strictly forbids making such data freely available. However, they can be obtained upon request from the JUMODA steering committee. Readers may contact: diane.korb@inserm.fr or epope@inserm.fr to request the data.

5. One of the noted authors is a group or consortium: JUmeaux MODe d’Accouchement (JUMODA) study group and the Groupe de Recherche en Obstétrique et Gynécologie (GROG)

In addition to naming the author group, please list the individual authors and affiliations within this group in the acknowledgments section of your manuscript. Please also indicate clearly a lead author for this group along with a contact email address.

Individual collaborators of JUMODA group and GROG are not co-authors of this manuscript.

We have added in the acknowledgments section the individual collaborators of JUMODA group and GROG and their affiliations. In supplementary file, we added the lead author for these groups along with contact email address.

Response to the Reviewers

Reviewer #1: This manuscript uses a large population-based study of twin pregnancies "Jumonda" to identify risk factors for severe maternal morbidity and was planned prospectively. The primary outcome was a composite of including maternal death, severe hemorrhage (defined by need for second line therapy), transfusion >=units of pRBCs, uterine artery embolization, and several other severe adverse outcomes. They identified two subgroups of women with a >25% risk of developing severe acute maternal morbidity: those with both oocyte donation, non-severe preeclampsia, and nulliparas born in Sub-Saharan Africa with non-severe preeclampsia.

We thank the Reviewer for his/her comment.

Reviewer #2: 

1. This is a potential wonderful paper. I am afraid I have major problems with the definition of severe morbidity.

Pre ecclampsia requiring delivery is certainly serious but NOT always a severe morbid event.

We agree with the Reviewer that preeclampsia can reflect different steps within the morbidity continuum of hypertension-related complications. 

The definition of severe preeclampsia in our study was the one retained in the EPIMOMS study. The EPIMOMS definition of severe maternal morbidity was obtained via an extensive national Delphi expert consensus process, with a panel of professionals including obstetricians, midwives, anaesthesiologists/intensive care specialists, and public health specialists. The process followed a Delphi-Rand design and consisted of two rounds followed by a final plenary session. The level of consensus for each round was fixed at 70%. Several published analyses were conducted with this definition confirming the recognition of its validity. (1-5)

In this multicriteria definition of severe maternal morbidity, severe preeclampsia can be defined through various criteria that indicate maternal : either a preeclampsia which indicated an induction of a preterm delivery before 32 gestational weeks for a maternal indication; or a preeclampsia associated with eclampsia, HELLP syndrome, or placental abruption; or responsible for a severe organ dysfunction (as per Epimoms specific definitions for each dysfunction): respiratory dysfunction, renal dysfunction, hepatic dysfunction. 

To define severe (for the mother) preeclampsia apart from the aforementioned preeclampsia complications, criterion retained was the gestational age at induced delivery and not proteinuria or blood pressure cut-offs, because thresholds are not consensual. Severe preeclampsia remote from term is challenging and requires assessment of the benefit-risk balance for the fetus and for the mother between an expectant management and an induced preterm delivery. Before 32 gestational weeks if an induced preterm delivery was decided for a main maternal indication, this can only be for a severe event compromising maternal health allowing to consent for neonatal morbidity induced by prematurity.

Furthermore, we performed a sensitivity analysis after exclusion of severe acute maternal morbidity cases only due to hypertensive complications (n= 19 women) and our results were unchanged (S4A Table), as mentioned in the methods section line 137 to 142 and in the results section line 247 to 252.

Therefore, we believe that SAMM events due to severe preeclampsia were always severe morbid events and do not compromise the validity of our results. 

2. In a large network Criteria to admit to an ICU will not all be the same and thus potential for bias.

We agree with the Reviewer that intensive care units (ICU) admission can depend on several factors, including the organization of the health care system and the accessibility and availability of ICU. (6,7) However, maternal admission to ICU has been considered by many authors as a ‘‘proxy’’ for reasonably assessing SAMM. (8-16) In France, rates, causes and severity of maternal admissions to intensive care units were analyzed in a recent study and showed a decrease in the rate of maternal ICU admissions from 2010 to 2014, but that the admitted mothers presented more severe clinical conditions, requiring more often resuscitation procedures and techniques provided in ICU. (17) These results suggest that admission to ICU increasingly reflects severe maternal morbidity.

Moreover in our study, among the 121 women admitted to ICU, only 16 did not have any other criterion of the composite definition of SAMM, and indications of admission to ICU confirm the severity of this event (table below).

Therefore, we do not believe that inclusion of the ICU admission criterion could have resulted in a potential bias.

Table: Indications of admission to ICU in JUMODA study

Indications of admission to ICU (not exclusive) N (%)

N=121

Severe postpartum hemorrhage 75 (62.0)

Eclampsia 5 (4.1)

HELLP syndrom 24 (19.8)

Other hypertensive complications 7 (5.8)

Respiratory dysfunction 5 (4.1)

Severe infection 5 (4.1)

Pulmonary embolism 3 (2.5)

Hepatic dysfunction 3 (2.5)

Others 10 (8.3)

3. This has the potential for a significant impact please try to find another channel is Tiffany and why except it if I am open even though yours was conducted using a a delfi

Can you be more strict in your defn of SMM and redo ? It will be more meaningful for everyone

For the reasons provided in response to previous comments, we believe that the Epimoms definition of SAMM obtained through a national consensus of experts is valid to define events of severe maternal morbidity and that it is not necessary to reanalyze the data with a more strict definition of SAMM.

Our results will be unchanged with a definition of SAMM excluding preeclampsia and admission to ICU. 

References : 

(1) Siddiqui A, Azria E, Howell EA, Deneux-Tharaux C. Associations between maternal obesity and severe maternal morbidity: Findings from the French EPIMOMS population-based study. Paediatr Perinat Epidemiol. 2019 Jan;33:7-16

(2) Korb D., Goffinet A., Seco A., Chevret S., Deneux-Tharaux C., pour le groupe EPIMOMS,Risk of severe maternal morbidity associated with cesarean delivery and the role of maternal age: a population-based propensity score analysis. CMAJ 2019 April 1;191:E352-60

(3) Madar H, Goffinet F, Seco A, Rozenberg P, Dupont C, Deneux-Tharaux C. Severe Acute Maternal Morbidity in Twin Compared With Singleton Pregnancies. Obstet Gynecol. 2019 Jun;133:1141–50.

(4) Korb D., Deneux-Tharaux C., Seco A., Goffinet F., Schmitz T., Risk of Severe Acute Maternal Morbidity According to Planned Mode of Delivery in Twin Pregnancies. Obstet Gynecol 2018, 132, 647-655

(5) Le Ray C, Pelage L, Seco A, Bouvier-Colle M-H, Chantry AA, Deneux-Tharaux C. Risk of severe maternal morbidity associated with in vitro fertilization: a population-based study. BJOG Int J Obstet Gynaecol. 2019 Jul;126:1033-1041

(6) Chantry AA, Deneux-Tharaux C, Bonnet MP, Bouvier-Colle MH. Pregnancy related

ICU admissions in France: trends in rate and severity, 2006-2009. Crit Care Med 2015;43:78–86.

(7) DREES. La Statistique annuelle des etablissements (SAE), http://drees.solidarites-

sante.gouv.fr/etudes-et-statistiques/open-data/etablissements-desante-

sociaux-et-medico-sociaux/article/la-statistique-annuelle-des-etablissements-

sae

(8) Baskett TF, O’Connell CM. Severe obstetric maternal morbidity: a 15-year population-based study. J Obstet 2005;25:7–9. 

(9) Madan I, Puri I, Jain NJ, Grotegut C, Nelson D, Dandolu V. Characteristics of obstetric intensive care unit admissions in New Jersey. JMatern Fetal Neonatal Med 2009;22:785–90.

(10) Pollock W, Rose L, Dennis CL. Pregnant and postpartum admissions to the intensive care unit: a systematic review. Intensive CareMed 2010;36:1465–74.

(11) Reichenheim ME, Zylbersztajn F, Moraes CL, Lobato G. Severe acute obstetric morbidity (near-miss): a review of the relative use of its diagnostic indicators. Arch Gynecol Obstet 2009;280:337–43.

(12) Senanayake H, Dias T, Jayawardena A. Maternal mortality and morbidity: epidemiology of intensive care admissions in pregnancy. Best Pract Res Clin Obstet 2013;27:811–20.

(13) Tuncalp O, Hindin MJ, Souza JP, Chou D, Say L. The prevalence of maternal near miss: a systematic review. BJOG 2012;119:653–61.

(14) Wanderer JP, Leffert LR, Mhyre JM, Kuklina EV, Callaghan WM, Bateman BT. Epidemiology of obstetric-related ICU admissions in Maryland: 1999–2008*. Crit Care Med 2013;41:1844–52.

(15) Zeeman GG. Obstetric critical care: a blueprint for improved outcomes. Crit Care Med 2006;34:S208–14.

(16) Zwart JJ, Dupuis JR, Richters A, Ory F, van Roosmalen J. Obstetric intensive care unit admission: a 2-year nationwide population-based cohort study. Intens Care Med 2010;36:256–63.

(17) Barry Y, Deneux-Tharaux C, Saucedo M, Goulet V, Guseva-Canu I, Regnault N, et al. Maternal admissions to intensive care units in France: Trends in rates, causes and severity from 2010 to 2014. Anaesth Crit Care Pain Med. 2019 Aug;38:363–9.

---

## [Decision Letter · Decision Letter 1]

11 Feb 2020

Risk factors and high-risk subgroups of severe acute maternal morbidity in twin pregnancy: a population-based study

PONE-D-19-27799R1

Dear Dr. Korb,

We are pleased to inform you that your manuscript has been judged scientifically suitable for publication and will be formally accepted for publication once it complies with all outstanding technical requirements.

With kind regards,

Pal Bela Szecsi, M.D. D.M.Sci.

Academic Editor

PLOS ONE

Additional Editor Comments (optional):

Reviewers' comments:

Reviewer's Responses to Questions

**Comments to the Author**

1. If the authors have adequately addressed your comments raised in a previous round of review and you feel that this manuscript is now acceptable for publication, you may indicate that here to bypass the “Comments to the Author” section, enter your conflict of interest statement in the “Confidential to Editor” section, and submit your "Accept" recommendation.

Reviewer #2: All comments have been addressed

Reviewer #3: All comments have been addressed

2. Is the manuscript technically sound, and do the data support the conclusions?

Reviewer #2: Yes

Reviewer #3: Yes

3. Has the statistical analysis been performed appropriately and rigorously? 

Reviewer #2: Yes

Reviewer #3: Yes

4. Have the authors made all data underlying the findings in their manuscript fully available?

Reviewer #2: Yes

Reviewer #3: No

5. Is the manuscript presented in an intelligible fashion and written in standard English?

Reviewer #2: Yes

Reviewer #3: Yes

6. Review Comments to the Author

Reviewer #2: (No Response)

Reviewer #3: Thank you very much for giving me the opportunity to review the revised version of this manuscript.

The current prospective study regards the evaluation of SAMM in twin pregnancy and explore the outcomes identifying relevant subgroups at major risk.

The authors approached the research question correctly from a methodological point of view, defining the so-called basal risk and the subsequent risks that occurred during pregnancy. I think that CART analysis approach and, the subsequent sensitive analysis, represent an optimal approach to the question of the authors. Moreover, the results of the study confirmed a "common-feeling" of increased risk in HRT twin pregnancies.

The authors clearly comment on the strengths and limitations of the study. So far, I agree with the response of the authors regarding the criticism of admissions to ICU and the improbability of a potential selection bias and I agree as well to the other criticism raised and relative responses of the authors.

In my opinion this is a precious paper that has an immediate clinical implication for an active increased surveillance in the subgroups identified.

I have no major concerns on this paper.

7. PLOS authors have the option to publish the peer review history of their article (what does this mean?). If published, this will include your full peer review and any attached files.

Reviewer #2: No

Reviewer #3: No

---

## [Editor Report · Acceptance letter]

14 Feb 2020

PONE-D-19-27799R1 

Risk factors and high-risk subgroups of severe acute maternal morbidity in twin pregnancy: a population-based study 

Dear Dr. Korb:

I am pleased to inform you that your manuscript has been deemed suitable for publication in PLOS ONE. Congratulations! Your manuscript is now with our production department. 

With kind regards,

on behalf of

Dr. Pal Bela Szecsi 

Academic Editor

PLOS ONE